# Solvophobicity-directed assembly of microporous molecular crystals

Hiroshi Yamagishi [1✉], Monika Tsunoda[1], Kohei Iwai[1], Kowit Hengphasatporn[2], Yasuteru Shigeta [2], Hiroyasu Sato[3] & Yohei Yamamoto [1]

Dense packing is a universal tendency of organic molecules in the solid state. Typical porous crystals utilize reticular strong intermolecular bonding networks to overcome this principle. Here, we report a solvophobicity-based methodology for assembling discrete molecules into a porous form and succeed in synthesizing isostructural porous polymorphs of an amphiphilic aromatic molecule $Py_6Mes$. A computational analysis of the crystal structure reveals the major contribution of dispersion interaction as the driving force for assembling $Py_6Mes$ into a columnar stacking while the columns are sterically salient and form nanopores between them. The porous packing is facilitated particularly in solvents with weak dispersion inter-action due to the solvophobic effect. Conversely, solvents with strong dispersion interaction intercalate between $Py_6Mes$ due to the solvophilic effect and provide non-porous inclusion crystals. The solvophobicity-directed polymorphism is further corroborated by the poly-morphs of $Py_6Mes$-analogues, $m$-$Py_6Mes$ and $Ph_6Mes$.

[1] Department of Materials Science, Faculty of Pure and Applied Sciences, and Tsukuba Research Center for Energy Materials Science (TREMS), University of Tsukuba, Tsukuba, Ibaraki, Japan. [2] Center for Computational Sciences, University of Tsukuba, Tsukuba, Ibaraki, Japan. [3] Rigaku Corporation, Akishima, Tokyo, Japan. ✉email: yamagishi.hiroshi.ff@u.tsukuba.ac.jp

Organic molecules tend to form a dense crystal with minimal void volume so that the molecules therein can maximize the intermolecular interactions between the adjacent molecules[1–3]. The synthesis of a porous crystal thus requires a tailored molecular design to overcome this universal tendency. To this end, established porous crystals, such as metal–organic frameworks, typically employ organic linkers featuring multiple adhesive functional groups that can bind with each other to form a reticular framework[4–8].

A fundamental question here is whether it is really unfeasible to assemble nonfunctional discrete molecules in a sparse manner. Although this question appears contradictory to the above-described tendency toward dense packing, there actually exist a handful of successful examples[9–20]. Organic zeolites are a well-known class of such compounds that can uptake/release guest solvent molecules efficiently, and selectively depending on the geometry and chemical affinity, yet organic zeolites are not truly porous materials because their pores readily collapse upon removing the guests[21–24]. More recently, several organic crystals that can retain vacant pores have been developed[9–20]. These compounds spontaneously assemble into a porous packing despite the fact that the packing is sustained only by weak interactions, including C–H···X bonds, π–π stacking, halogen bonds, and van der Waals (vdW) forces, whose bonding strength are much less than the conventional hydrogen bonding ($15$–$60\,\text{kJ mol}^{-1}$)[25]. These porous molecular crystals are intriguing not only fundamentally but also practically because of their distinct solution processability, structural flexibility, and self-healing ability, which are largely prohibited in the conventional porous crystals[17–20]. However, it still remains unexplored how one can drive the discrete molecules to assemble sparsely[1,26]. Moreover, with the existing compounds, crystallization solvent and procedure have to be carefully designed. Otherwise, the porous molecular crystals readily collapse into a densely packed polymorph, which further prohibits their development. In fact, most of the reported stable porous molecular crystals were found by chance except those composed of intrinsically porous molecular cages[27–29].

Previously, we reported a porous molecular crystal $\mathbf{Py^{open}{\cdot}MeCN}$ composed of a $D_{3h}$-symmetric amphiphilic aromatic compound $\mathbf{Py_6Mes}$ (Fig. 1a)[18,20]. $\mathbf{Py_6Mes}$ assembled together via multiple C–H···N bonds to form a molecular framework with one-dimensional micropores (Fig. 2a), which can maintain its porous architecture up to 202 °C. Although further heating ended up with

the collapse of the pores, the resultant nonporous polymorph $\mathbf{Py^{close}}$ spontaneously self-healed back into $\mathbf{Py^{open}{\cdot}MeCN}$ upon exposure to vapor of MeCN at ambient temperature. We anticipated that $\mathbf{Py^{open}{\cdot}MeCN}$, featuring excellent thermal stability and compositional simplicity, could serve as a highly promising platform for investigating how discrete molecules assemble into a porous form. Along this line, we also reported, in the previous report, a plausible molecular assembly mechanism for $\mathbf{Py^{open}}$ based on its four types of polymorphs[18]. However, the available crystallographic data were limited at that period and, thus, we could not establish a reliable and general design strategy toward porous molecular crystals.

Here, we report a molecular strategy for synthesizing isomorphic porous molecular crystals from various organic solvents. Through a detailed computational investigation, we reveal the major contribution of dispersion force in the assembling process of the constituent $\mathbf{Py_6Mes}$ molecules into a porous manner. Following this understanding, we crystalize $\mathbf{Py_6Mes}$ and succeed in synthesizing porous polymorphs in solvents with less dispersion interaction due to the solvophobic effect (Fig. 1b). In contrast, solvents with larger dispersion interaction provide nonporous inclusion crystals due to the solvophilic effect. Newly synthesized $\mathbf{Py_6Mes}$ analogs, $\mathbf{\textit{m}\text{-}Py_6Mes}$ (Fig. 1a) and $\mathbf{Ph_6Mes}$, also show consistent solvophobicity-directed polymorphism.

## Results and discussion

**Energy decomposition analysis of $\mathbf{Py^{open}{\cdot}MeCN}$.** To gain insight into how a discrete molecule assembles into a porous form, we focus on $\mathbf{Py^{open}{\cdot}MeCN}$, whose crystal structure was previously identified[18]. In $\mathbf{Py^{open}{\cdot}MeCN}$, $\mathbf{Py_6Mes}$ stacks with each other to form a one-dimensional column along the twofold screw axis (crystallographic $b$-axis, Fig. 2c). The polar pyridine rings and the nonpolar benzene and mesitylene rings are spatially segregated in the column to form a polar shell and a nonpolar interior (Fig. 2b). We conduct a computational calculation of the intermolecular interactions between the adjacent $\mathbf{Py_6Mes}$ molecules in $\mathbf{Py^{open}{\cdot}MeCN}$. Pair interaction energy decomposition analysis (PIEDA)[30] is performed for this purpose by using the fragment molecular orbital (FMO)[31] method at the RI-MP2 level of theory with $6\text{-}31+\text{G(d)}$ basis set (Supplementary Fig. 23 and Supplementary Table 15, see "Methods" for the details of the computational methods). A negative value represents an attractive interaction. $E^{\text{vdW}}$, $E^{\text{ES}}$, and $E^{\text{CT}+\text{mix}}$ respectively represent

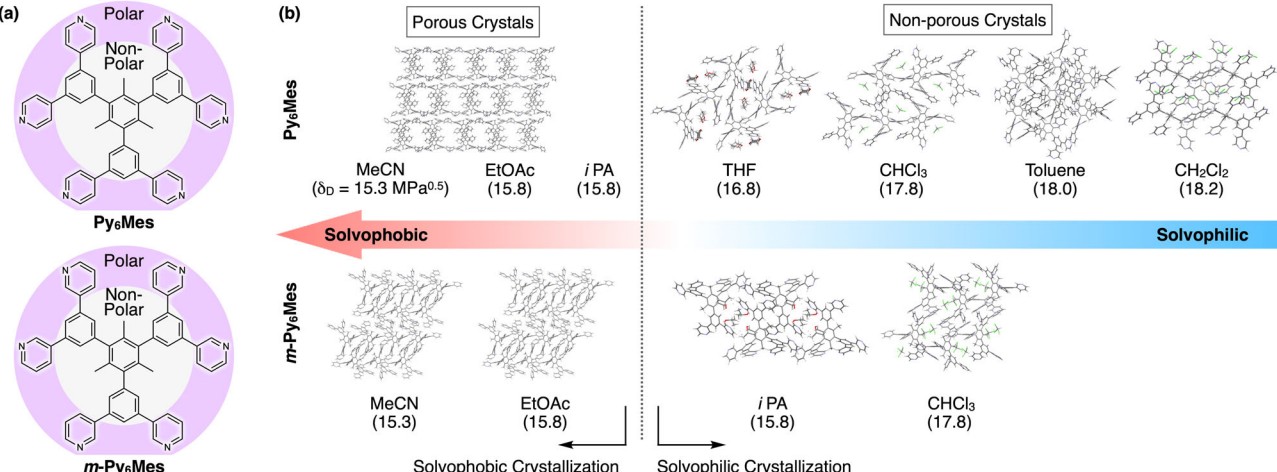

**Fig. 1 Schematic representations of the $\delta_D$-dependent polymorphism of $\mathbf{Py_6Mes}$ and $\mathbf{\textit{m}\text{-}Py_6Mes}$. a** Molecular structures of $\mathbf{Py_6Mes}$ and $\mathbf{\textit{m}\text{-}Py_6Mes}$ with their polar peripheries and nonpolar cores highlighted in violet and gray. **b** Crystal packing diagrams of polymorphs of $\mathbf{Py_6Mes}$ and $\mathbf{\textit{m}\text{-}Py_6Mes}$. Hansen dispersion cohesion parameters ($\delta_D$) of the crystallization solvents are given in the parentheses.

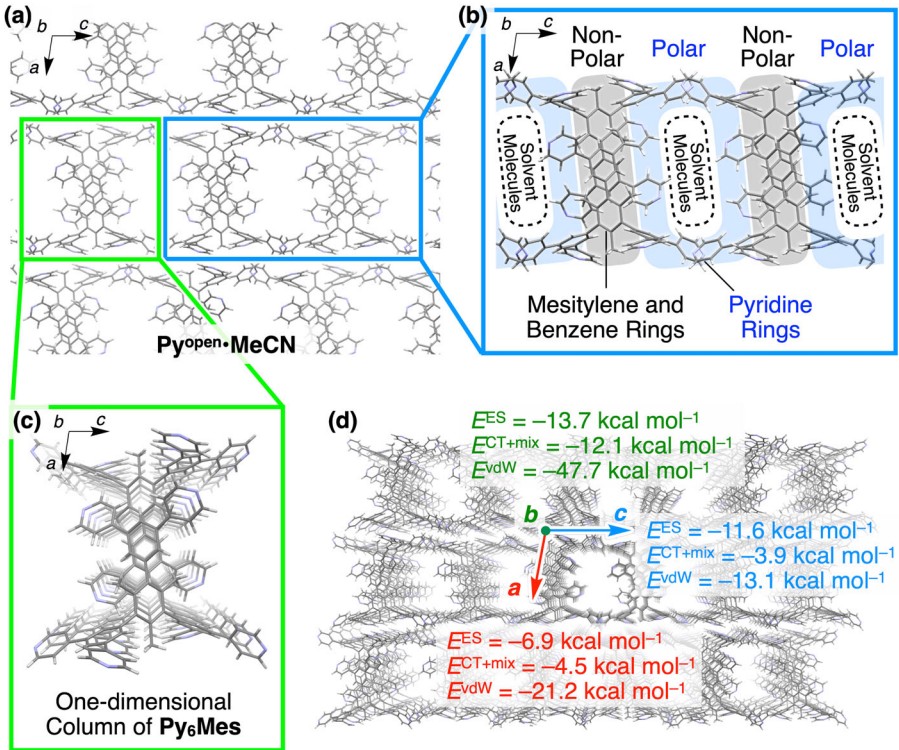

**Fig. 2 Computational analysis of the attractive and repulsive energies in Py$^{open}$·MeCN. a** A crystal packing diagram of **Py$^{open}$·MeCN** viewed along the crystallographic *b*-axis. **b** A partial crystal packing of **Py$^{open}$·MeCN** highlighting the polar shell and nonpolar core. **c** A columnar stacking of **Py$_6$Mes** in **Py$^{open}$·MeCN**. **d** Electrostatic energy ($E^{ES}$), charge transfer energy with higher-order mixed terms energies ($E^{CT+mix}$), and dispersion energy ($E^{vdW}$) exserted along the crystallographic *a*- (red), *b*- (green), and *c*-axes (blue) in **Py$^{open}$·MeCN**.

the vdW dispersion energy, electrostatic energy, and charge transfer energy with higher-order mixed term energies. Despite the sparse and porous structure, the overall interaction between **Py$_6$Mes** is relatively large (−94.3 kcal mol$^{-1}$, Supplementary Table 15), explaining the excellent thermal stability of **Py$^{open}$·MeCN**. The prominent energetic gain along the crystallographic *b*-axis (Fig. 2d) indicates the preferential formation of the columnar stacking of **Py$_6$Mes** (Fig. 2c) at the expense of the energetic gain obtained from the intercolumnar packing along the crystallographic *a*- and *c*-axes. As expected from the richness of C–H⋯N bond, dispersion interaction is the major attractive contribution in the crystal, especially along the crystallographic *a*- and *b*-axes (Fig. 2d). Electrostatic interaction as well as dispersion interaction is prominent along the crystallographic *c* axis (Fig. 2d). Altogether, dispersion interaction occupies 60.9% of the total attractive energy of −134.3 kcal mol$^{-1}$ in **Py$^{open}$·MeCN** (Supplementary Table 15).

Subsequently, we conducted computational investigation into the effect of polarity of the crystallization solvents, which has been considered as an essential parameter for predicting the polymorphism. We calculate the total system energy of **Py$^{open}$** on the assumption that the constituent **Py$_6$Mes** molecules are surrounded by MeOH, CHCl$_3$, acetone, toluene, and dichloroethane, respectively, which are available in GAMESS program[32]. As summarized in Supplementary Table 16, the porous architecture is stabilized more as the polarity of the surrounding solvent increases, yet the change in stabilization energy from the surrounding environment estimated by the polarized continuum model method is relatively small in comparison with the energetic gain from dispersion force. Overall, the porous assembly of **Py$_6$Mes** is sustained dominantly by the dispersion forces together with the stabilization by the polarity of the surrounding media.

**Crystallographic analysis of polymorphs of Py$_6$Mes.** Based on the understanding obtained from the calculation, we crystalize **Py$_6$Mes** from a series of common organic solvents, and analyze their crystal structures with the aim to control the intra- and intercolumnar stacking of **Py$_6$Mes**. As a typical recrystallization procedure, saturated solution of **Py$_6$Mes** is poured into a small glass vial, which is loosely sealed with a cap and stood at 25 °C for several days to allow the mother solvent to sluggishly evaporate. In the previous report, we utilized MeCN, 2-propanol (*i*PA), tetrahydrofuran (THF), and CHCl$_3$ as the crystallization solvents of **Py$_6$Mes**. Here, we additionally utilize MeOH, EtOH, butyronitrile (BN), EtOAc, acetone, 1-chloropropane (PrCl), 1-butanol (BuOH), toluene, CH$_2$Cl$_2$, dimethylsulfoxide (DMSO), and *γ*-butyrolactone (GBL) as the crystallization solvents.

Some of the non-protic solvents (EtOAc, CH$_2$Cl$_2$, and toluene) successfully give crystalline precipitates of **Py$_6$Mes** that are applicable for the single-crystal X-ray diffraction structure analysis. Crystallographic information and the symbols of the resultant crystals are summarized in Table 1 and Supplementary Tables 1–3. Highly polar protic solvents (MeOH and EtOH) are inappropriate for the crystallization due to their poor solubility. Other solvents yield fine crystalline powders, which are analyzed by PXRD.

The single crystal obtained from EtOAc (**Py$^{open}$·EtOAc**) features a porous molecular packing that is virtually identical with **Py$^{open}$·MeCN** and **Py$^{open}$·iPA** (Fig. 2a and Supplementary Fig. 9). Pore size distribution of **Py$^{open}$·EtOAc** calculated from its N$_2$ adsorption isotherm profile (Supplementary Fig. 22) is nearly identical with that of **Py$^{open}$·MeCN**[18], while its BET surface area (597 m$^2$ g$^{-1}$) is larger than **Py$^{open}$·MeCN** plausibly due to the higher structural integrity of **Py$^{open}$·EtOAc** crystals. Trials to assign the guest solvent molecules trapped inside the pore are

**Table 1 Crystallographic information of the polymorphs of Py$_6$Mes.**

| Crystallization solvent | Symbol | Space group | $\varepsilon$ | $\delta_D$ (MPa$^{0.5}$) | Cell volume (Å$^3$) | Volume per Py$_6$Mes (Å$^3$) | C–H···N per Py$_6$Mes | C–H···π per Py$_6$Mes | Sum of the contacts |
|---|---|---|---|---|---|---|---|---|---|
| MeCN | Py$^{open}$•MeCN | P2$_1$/c | 37.5 | 15.3 | 5273 | 1318 | 5 | 2 | 7 |
| EtOAc | Py$^{open}$•EtOAc | P2$_1$/c | 6.02 | 15.8 | 5404 | 1351 | 6 | 1 | 7 |
| iPA | Py$^{open}$•iPA | P2$_1$/c | 18.3 | 15.8 | 5278 | 1320 | 5 | 2 | 7 |
| THF | Py$^{VDW}$•THF | P2$_1$/n | 7.52 | 16.8 | 5252 | 1313 | 1 | 1 | 2 |
| CHCl$_3$ | Py$^{VDW}$•CHCl$_3$ | P2$_1$/n | 4.81 | 17.8 | 4983 | 1246 | 2 | 2 | 4 |
| Toluene | Py$^{VDW}$•C$_7$H$_8$ | P-1 | 2.38 | 18.0 | 4872 | 1218 | 4 | 2 | 6 |
| CH$_2$Cl$_2$ | Py$^{VDW}$•CH$_2$Cl$_2$ | C2/c | 9.08 | 18.2 | 4813 | 1203 | 4 | 0 | 4 |

The space groups, cell volumes, and the number of egoistic C–H···N and C–H···π contacts per one **Py$_6$Mes** molecule found in **Py$^{open}$·MeCN**[18], **Py$^{open}$·EtOAc**, **Py$^{open}$·iPA**[18], **Py$^{VDW}$·THF**[18], **Py$^{VDW}$·CH$_2$Cl$_2$**, **Py$^{VDW}$·CHCl$_3$**[18], and **Py$^{VDW}$·C$_7$H$_8$** together with the sum of the contacts. The relative permittivity ($\varepsilon$)[37] and the Hansen dispersion cohesion parameters ($\delta_D$)[34] of the crystallization solvents are also listed.

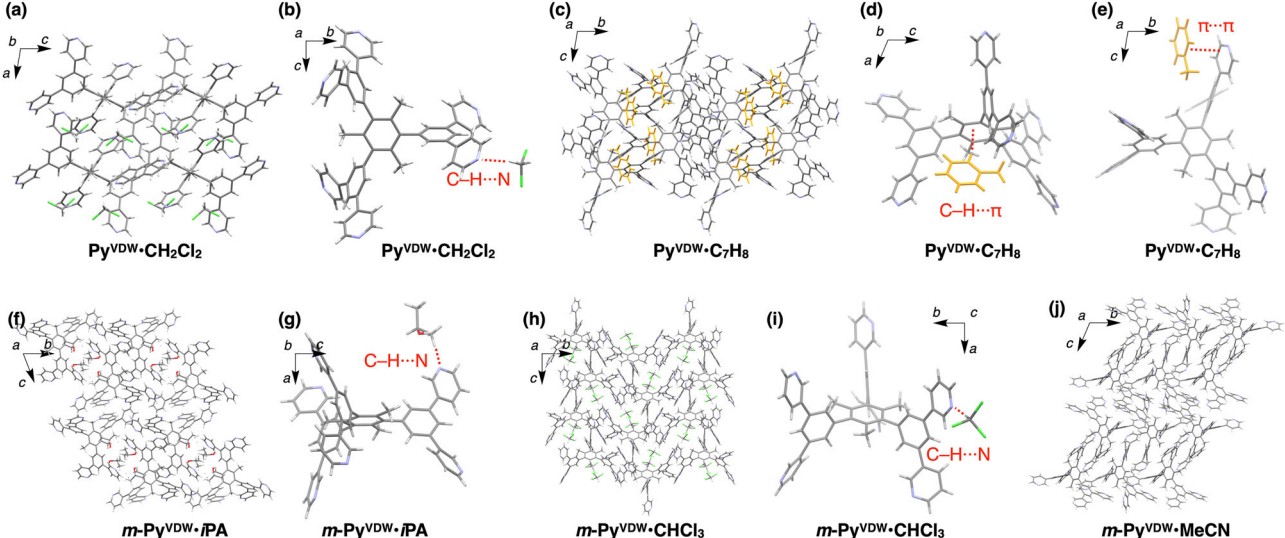

**Fig. 3 Crystal packing diagrams of the polymorphs of Py$_6$Mes and m-Py$_6$Mes. a, c, f, h, j** Crystal packing diagrams of **Py$^{VDW}$·CH$_2$Cl$_2$** (**a**), **Py$^{VDW}$·C$_7$H$_8$** (**c**), **m-Py$^{VDW}$·iPA** (**f**), **m-Py$^{VDW}$·CHCl$_3$** (**h**), and **m-Py$^{VDW}$·MeCN** (**j**). **b, d, e, g, i** Partial crystal structures of **Py$^{VDW}$·CH$_2$Cl$_2$** (**b**), **Py$^{VDW}$·C$_7$H$_8$** (**d,e**), **m-Py$^{VDW}$·iPA** (**g**), and **m-Py$^{VDW}$·CHCl$_3$** (**i**). The solvent–solute interactions are visualized with red dashed lines. The guest toluene molecules are colored in orange for clarity.

unsuccessful for all the porous crystals obtained from MeCN, iPA, and EtOAc (**Py$^{open}$·MeCN**, **Py$^{open}$·iPA**, and **Py$^{open}$·EtOAc**) due to the severe disorder. Some residual electron density is detected in the pores according to the SQUEEZE program[33] (66, 52, and 53 electrons for MeCN, iPA, and EtOAc, respectively), which indicates the inclusion of certain amount of the crystallization solvent molecules in the pores.

Inclusion molecular crystals **Py$^{VDW}$·CH$_2$Cl$_2$** and **Py$^{VDW}$·C$_7$H$_8$** are obtained, respectively, from CH$_2$Cl$_2$ and toluene. **Py$^{VDW}$·CH$_2$Cl$_2$** (Fig. 3a and Supplementary Fig. 10) belongs to a space group of C2/c, in which eight non-disordered CH$_2$Cl$_2$ molecules pack together with four molecules of **Py$_6$Mes** in a unit cell. The H atoms in CH$_2$Cl$_2$ make a short contact with a N atom in **Py$_6$Mes** (2.594 Å; Fig. 3b). **Py$_6$Mes** molecules form C–H···N contacts (2.583 and 2.500 Å) with each other along with several C–H···H contacts. **Py$^{VDW}$·C$_7$H$_8$** (Fig. 3c and Supplementary Fig. 11) belongs to a space group of P-1, in which four toluene molecules pack together with four **Py$_6$Mes** molecules in a unit cell. The guest toluene molecules form C–H···π contacts and a π–π stacking with **Py$_6$Mes** (Fig. 3d, e). **Py$_6$Mes** molecules form eight C–H···N contacts with each other along with C–H···π contacts and π–π stackings.

Precipitates obtained in other solvents are analyzed by PXRD due to the difficulty in synthesizing diffraction-quality single

crystals (Supplementary Fig. 8). BN solution of **Py$_6$Mes** affords a porous crystal that is isomorphic to **Py$^{open}$**, while the crystals obtained from other solvents (acetone, PrCl, BuOH, DMSO, and GBL) are not isomorphic to **Py$^{open}$**. The single-crystal structures, PXRD profiles, and the physical properties of the crystallization solvents (relative permittivity and Hansen parameters[34]) are summarized in Supplementary Table 14 along with those of mesitylene, benzene, and pyridine as the components of **Py$_6$Mes**.

It is worth noting that isomorphic porous crystals were obtained from a variety of solvents (MeCN, iPA, BN, and EtOAc) that are seemingly irrelevant with each other, in terms of the polarity or hydrogen bonding capability. This is in clear contrast with the previously reported porous molecular crystals that were basically sensitive to the crystallization solvents or crystallization procedures[1].

**Hansen solubility parameter and polymorphism.** Relative permittivity ($\varepsilon$) has often been regarded as the primal parameter for the prediction of the solute–solvent interactions. However, the tendency in polymorphism of **Py$_6$Mes** toward $\varepsilon$ is indistinct (Supplementary Table 14). We presume that, based on the results from the computational analysis, the capability of forming the dispersion interaction may govern the polymorphism. To prove

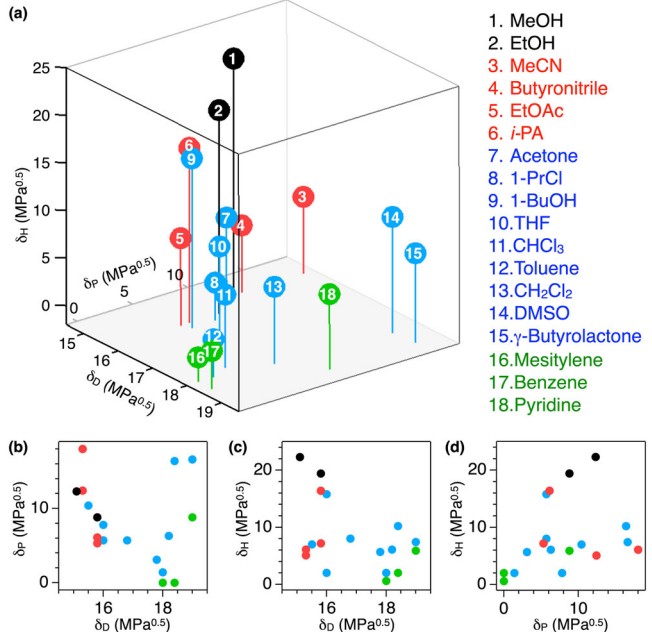

(a)
1. MeOH
2. EtOH
3. MeCN
4. Butyronitrile
5. EtOAc
6. i-PA
7. Acetone
8. 1-PrCl
9. 1-BuOH
10. THF
11. CHCl₃
12. Toluene
13. CH₂Cl₂
14. DMSO
15. γ-Butyrolactone
16. Mesitylene
17. Benzene
18. Pyridine

**Fig. 4 Hansen space for polymorphs of Py₆Mes. a** Hansen space showing the polymorphism of **Py₆Mes**. The solvents that poorly dissolve **Py₆Mes**, and that afford porous crystals are respectively colored in black and red. The components of **Py₆Mes** are colored in green. The other solvents are colored in blue. **b–d** The projections of the Hansen space onto the $\delta_D\delta_P$- (**b**), $\delta_D\delta_H$- (**c**), and $\delta_P\delta_H$-planes (**d**).

this theory, we focus on Hansen parameters, which are the empirical values of the strength of dispersion ($\delta_D$), polar ($\delta_P$), and hydrogen bonding cohesion parameters ($\delta_H$). Besides the conventional utility for the prediction of the solubility of organic polymers, Hansen parameters have recently been applied for the prediction of polymorphism of some pharmaceutical molecules[35,36].

We apply this method to the polymorphism of **Py₆Mes**. The crystallization solvents and **Py₆Mes** components are plotted in the Hansen space according to their three coordinates of $\delta_D$, $\delta_P$, and $\delta_H$ (Fig. 4 and Supplementary Table 14). The **Py₆Mes** components (green spheres in Fig. 4a) feature large $\delta_D$ and relatively small $\delta_P$ and $\delta_H$. MeCN, BN, EtOAc, and iPA (red spheres in Fig. 4a) feature small $\delta_D$ and moderate or large $\delta_P$ and $\delta_H$. Highly polar solvents (MeOH and EtOH, black spheres in Fig. 4a) locate at the opposite corner from the **Py₆Mes** components. The other solvents (blue spheres in Fig. 4a) locate between the red and green spheres.

The geometrical distance in the Hansen space represents the solubility or affinity of given two substances. In line with this conventional understanding, the plot in Fig. 4a shows an explicit dependence on the distance from the **Py₆Mes** components. Solvents that are close to the **Py₆Mes** components yield nonporous polymorphs (blue spheres in Fig. 4a), while solvents that locate far from the **Py₆Mes** components are poor in solubility (black spheres in Fig. 4a). The remaining slightly affinitive solvents yield the porous crystals (red spheres in Fig. 4a).

This trend can be decomposed into the basic elements by focusing on the projections of the Hansen space onto the $\delta_D\delta_P$-, $\delta_D\delta_H$-, and $\delta_P\delta_H$-planes (Fig. 4b–d). As shown in Fig. 4d, the polymorphic tendency barely correlates with $\delta_P$ and $\delta_H$ of the crystallization solvents. On the other hand, $\delta_D$ describes the polymorphic tendency reasonably (Fig. 4b, c and Supplementary Table 14), namely, **Py₆Mes** crystalizes into the porous form when

the crystallization solvent can partially dissolve **Py₆Mes**, but is not affinitive with **Py₆Mes** especially in terms of the strength of the dispersion force.

We also analyze the intermolecular short contacts and crystal packing efficiency of the single crystals, with the aim to reveal the detailed solute–solvent interactions. The egoistic C–H···N and C–H···π contacts per one **Py₆Mes** molecule are summarized in Table 1. In the solvents with small $\delta_D$, **Py₆Mes** facilitates multiple C–H···N and C–H···π contacts with each other, while solvents with large $\delta_D$ suppress the egoistic contacts by intercalating between **Py₆Mes** molecules as shown in **Py$^{\text{VDW}}$·CH₂Cl₂** and **Py$^{\text{VDW}}$·C₇H₈** (Fig. 3a, c). At the same time, the inclusion of the solvent molecules optimizes the molecular packing of **Py₆Mes**. The averaged cell volume per one **Py₆Mes** molecule of the three porous crystals (**Py$^{\text{open}}$·MeCN**, **Py$^{\text{open}}$·EtOAc**, and **Py$^{\text{open}}$·iPA**) and the four inclusion crystals (**Py$^{\text{VDW}}$·THF**, **Py$^{\text{VDW}}$·CH₂Cl₂**, **Py$^{\text{VDW}}$·CHCl₃**, and **Py$^{\text{VDW}}$·C₇H₈**) are 1330 and 1245 Å³, respectively (Table 1). Namely, solvents with large $\delta_D$ are affinitive with **Py₆Mes** and allow the **Py₆Mes** to assemble into a dense packing by intercalating between **Py₆Mes** (solvophilic crystallization), while solvents with small $\delta_D$ are segregated from **Py₆Mes** due to the solvophobicity and facilitate the columnar assembly of **Py₆Mes** although the intercolumnar packing is not dense (solvophobic crystallization).

**Polymorphism of *m*-Py₆Mes and Ph₆Mes.** The $\delta_D$-dependent solvophilic/solvophobic crystallization is further corroborated by the polymorphs of ***m*-Py₆Mes** and **Ph₆Mes** (Figs. 1a, b and 5). Meta-substituted hexapyridyl mesitylene derivative ***m*-Py₆Mes** was newly synthesized by sequential Suzuki–Miyaura couplings of pyridineboronic acid, dibromo aniline, and triiodomesitylene (for details, see Supplementary Methods). Tri(terphenyl) mesitylene, **Ph₆Mes**, was newly synthesized by Suzuki–Miyaura coupling reaction of terphenylboronic acid and triiodomesitylene (for details, see Supplementary methods). The molecular structure of the resultant ***m*-Py₆Mes** and **Ph₆Mes** are unambiguously assigned by means of ¹H- and ¹³C-NMR spectroscopies, elemental analysis, and high-resolution mass spectrometry (Supplementary Figs. 1–7).

We crystalize ***m*-Py₆Mes** in the same way as **Py₆Mes** in MeCN, EtOAc, iPA, and CHCl₃ to obtain diffraction-quality single crystals. Their crystal packing diagrams and crystal structure information are shown in Fig. 3f–j and Table 2, respectively. Nonporous inclusion crystals are obtained when solvents with large $\delta_D$ (iPA and CHCl₃) are utilized (Fig. 3f, h, and Supplementary Figs. 14 and 15). The inclusion crystal obtained from iPA (***m*-Py$^{\text{VDW}}$·iPA**) belongs to a space group of P-1. The constituent ***m*-Py₆Mes** molecules form multiple C–H N and C–H···π contacts with each other along with a solvophilic C–H···N contact with a guest iPA molecule (Fig. 3g). The inclusion crystal with CHCl₃ (***m*-Py$^{\text{VDW}}$·CHCl₃**) belongs to a space group of P-1. The constituent ***m*-Py₆Mes** molecules form three C–H···N bonds with each other along with solvophilic C–H···N (Fig. 3i) and C–H···π contacts with guest CHCl₃ molecules.

The crystals obtained from MeCN (***m*-Py$^{\text{VDW}}$·MeCN**) and EtOAc (***m*-Py$^{\text{VDW}}$·EtOAc**) are isomorphic with each other, featuring no apparent pores or guest solvent molecules (Fig. 3j, and Supplementary Figs. 12 and 13). The crystal packing mode is basically analogous to the nonporous polymorph **Py$^{\text{close}}$**, which is obtained by thermal annealing of **Py$^{\text{open}}$** (see our previous report[18] for the detailed synthetic and structural information). In ***m*-Py$^{\text{VDW}}$·MeCN**, the constituent ***m*-Py₆Mes** molecules are solvophobically packed together to form five C–H···π contacts with each other in a unit cell. Moreover, 11 out of 12 pyridine

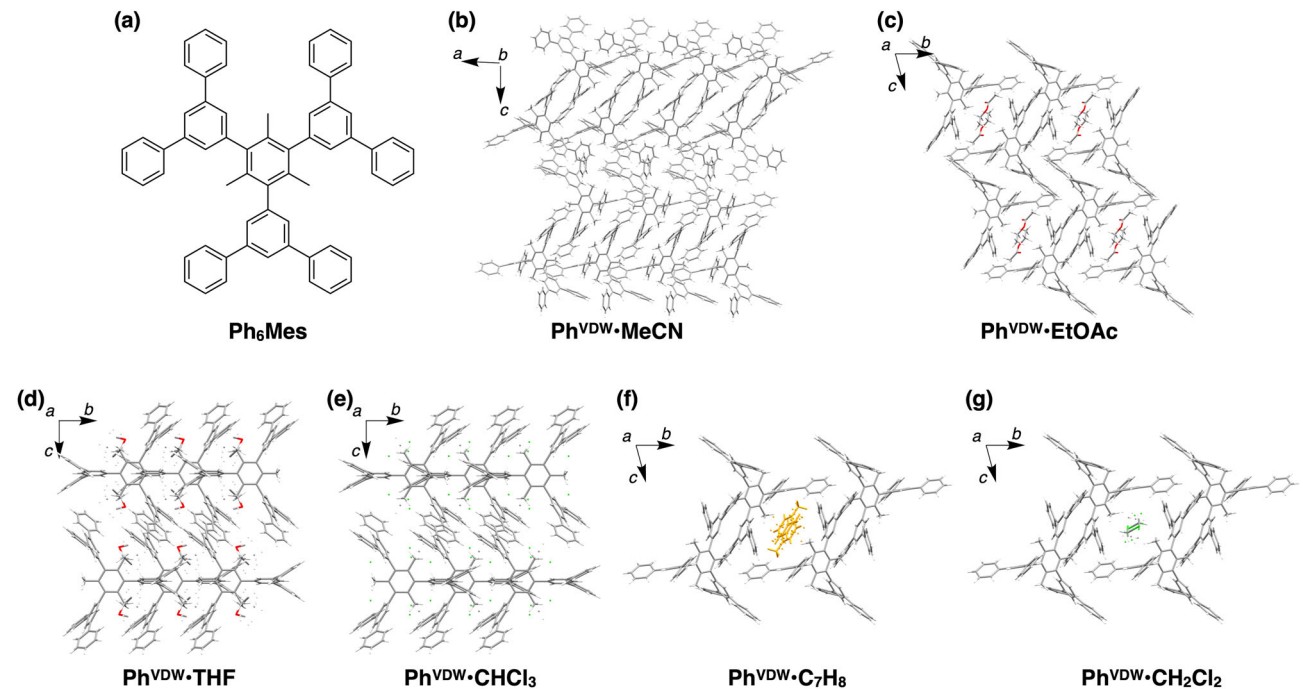

**Fig. 5 Crystal packing diagrams of the polymorphs of Ph$_6$Mes. a** Molecular structure of **Ph$_6$Mes**. **b–g** Crystal packing diagrams of **Ph$^{VDW}$·MeCN** (**b**), **Ph$^{VDW}$·EtOAc** (**c**), **Ph$^{VDW}$·THF** (**d**), **Ph$^{VDW}$·CHCl$_3$** (**e**), **Ph$^{VDW}$·C$_7$H$_8$** (**f**), and **Ph$^{VDW}$·CH$_2$Cl$_2$** (**g**). The guest toluene molecules are colored in orange for clarity.

**Table 2 Crystallographic information of the polymorphs of *m*-Py$_6$Mes.**

| Crystallization solvent | Symbol | $\varepsilon$ | $\delta_D$ (MPa$^{0.5}$) | Space group | Cell volume (Å$^3$) | Volume per Py$_6$Mes (Å$^3$) |
|---|---|---|---|---|---|---|
| MeCN | *m*-Py$^{VDW}$•MeCN | 37.5 | 15.3 | *P*-1 | 4274 | 1068 |
| EtOAc | *m*-Py$^{VDW}$•EtOAc | 6.02 | 15.8 | *P*-1 | 4205 | 1051 |
| iPA | *m*-Py$^{VDW}$•iPA | 18.3 | 15.8 | *P*-1 | 2552 | 1276 |
| CHCl$_3$ | *m*-Py$^{VDW}$•CHCl$_3$ | 4.81 | 17.8 | *P*2$_1$/*n* | 5079 | 1270 |

Crystal structure information of ***m*-Py$^{VDW}$·MeCN**, ***m*-Py$^{VDW}$·EtOAc**, ***m*-Py$^{VDW}$·iPA**, and ***m*-Py$^{VDW}$·CHCl$_3$** together with relative permittivity ($\varepsilon$)[37] and the Hansen dispersion cohesion parameters ($\delta_D$)[34] of the crystallization solvents.

**Table 3 Crystallographic information of the polymorphs of Ph$_6$Mes.**

| Crystallization solvent | Symbol | $\varepsilon$ | $\delta_D$ (MPa$^{0.5}$) | Space group | Cell volume (Å$^3$) | Volume per Py$_6$Mes (Å$^3$) |
|---|---|---|---|---|---|---|
| MeCN | Ph$^{VDW}$•MeCN | 37.5 | 15.3 | *P*-1 | 4348 | 1087 |
| EtOAc | Ph$^{VDW}$•EtOAc | 6.02 | 15.8 | *P*-1 | 2475 | 1238 |
| THF | Ph$^{VDW}$•THF | 7.52 | 16.8 | *C*2/*c* | 5207 | 1302 |
| CHCl$_3$ | Ph$^{VDW}$•CHCl$_3$ | 4.81 | 17.8 | *C*2/*c* | 5209 | 1302 |
| Toluene | Ph$^{VDW}$•C$_7$H$_8$ | 2.38 | 18.0 | *P*-1 | 2471 | 1236 |
| CH$_2$Cl$_2$ | Ph$^{VDW}$•CH$_2$Cl$_2$ | 9.08 | 18.2 | *P*-1 | 2464 | 1232 |

Crystal structure information of **Ph$^{VDW}$·MeCN**, **Ph$^{VDW}$·EtOAc**, **Ph$^{VDW}$·THF**, **Ph$^{VDW}$·CHCl$_3$**, **Ph$^{VDW}$·C$_7$H$_8$**, and **Ph$^{VDW}$·CH$_2$Cl$_2$** together with relative permittivity ($\varepsilon$)[37] and the Hansen dispersion cohesion parameters ($\delta_D$)[34] of the crystallization solvents.

rings in a unit cell form C–H···N bonds with the adjacent ***m*-Py$_6$Mes** molecules.

**Ph$_6$Mes** is analogously crystalized in MeCN, EtOAc, THF, CHCl$_3$, toluene and CH$_2$Cl$_2$, successfully yielding diffraction-quality single crystals, whose crystal packing diagrams and crystal structure information are shown in Fig. 5b–g, Table 3, Supplementary Figs. 16–21, and Supplementary Tables 8–13, respectively. In analogy with ***m*-Py$_6$Mes**, nonporous inclusion crystals are obtained when solvents with large $\delta_D$ (EtOAc, THF, CHCl$_3$, toluene, and CH$_2$Cl$_2$) are utilized (Fig. 5c–g and Supplementary

Figs. 17–21), while crystals from MeCN include no guest solvent molecules (Fig. 5b and Supplementary Fig. 16).

Polymorphs of ***m*-Py$_6$Mes** not only corroborate the $\delta_D$-dependency of **Py$_6$Mes** polymorphs, but also tell us about the delicate energetic balance between **Py$^{close}$** and **Py$^{open}$·MeCN**. Geometrically, **Py$_6$Mes** can assemble into a dense packing as proved by **Py$^{close}$**, ***m*-Py$^{VDW}$·MeCN**, or ***m*-Py$^{VDW}$·EtOAc**. However, unlike ***m*-Py$_6$Mes**, the position of the N atoms is static upon the rotation of the pyridyl rings around the single bond, which is unfavorable for the formation of multiple C–H···N bonds

with each other. Therefore, **Py₆Mes** may prefer to form a porous framework, in which **Py₆Mes** can form multiple C–H···N and C–H···π contacts with each other at the expense of the packing efficiency.

## Conclusion

In conclusion, we succeed in establishing a solvophobicity-based design strategy for the synthesis of porous molecular crystals and succeed in synthesizing porous molecular crystals by using various organic solvents. Energy decomposition analysis reveals the dominance of the dispersion energy as the attractive interaction in **Py$^{open}$·MeCN** especially in the columnar stacking, which is further stabilized by the polarity of the solvent. Consistently, solvents with small $\delta_D$ facilitate the egoistic assembly of **Py₆Mes** into a porous architecture via solvophobic interaction, while solvents with large $\delta_D$ intercalate between **Py₆Mes** via solvophilic interaction and provide nonporous inclusion polymorphs. The dominance of the dispersion energy as the attractive interaction in **Py$^{open}$·MeCN** is further supported by the polymorphism of ***m*-Py₆Mes** and **Ph₆Mes**. The combination of dispersion interaction as attractive force and solvophobicity as repulsive force, as presented in this paper, can be a conceptually novel strategy to go beyond the conventional porous crystal engineering that largely relies on the strong affinitive bonding networks.

## Methods

**Materials**. Commercial reagents were purchased from Sigma-Aldrich, TCI, and Wako Pure Chemical Industries, Ltd. All the chemicals are used as received unless otherwise mentioned.

**Reaction, purification, and characterization techniques**. All reactions were carried out under nitrogen atmosphere unless otherwise noted. Gel permeation column chromatography was performed on a Japan Analytical Industry model LC-9110 NEXT Recycling Preparative HPLC equipped with JAIGEL 2HH, by using CHCl₃ as eluent. ¹H and ¹³C NMR spectra were recorded on a JEOL model JNM-ECS-400 NMR spectrometer (¹H NMR, 400 MHz, ¹³C NMR, 100 MHz) JMTC-400/54/SS and a Bruker model AVANCE-600 NMR spectrometer (¹³C NMR, 150 MHz), using the residual solvent peak as an internal standard. High-resolution MS data were obtained using a Bruker model solariX XR Mass spectrometry in the positive mode with MeCN as solvent. Elemental analysis was conducted with an Elementar model organic elemental analyzer UNICUBE. The sorption isotherm measurement for N₂ (99.99995%) was performed using a Bel Japan, Inc. model BELSORP-max automatic volumetric adsorption apparatus. A known amount of **Py$^{open}$·EtOAc**, placed in a glass tube, was dried under a reduced pressure at 110 °C for 3 h to remove the included guest molecules.

**Typical procedure for the synthesis of single crystals of Py₆Mes, *m*-Py₆Mes, and Ph₆Mes**. A glass vial containing saturated solution of **Py₆Mes**, ***m*-Py₆Mes**, or **Ph₆Mes** was placed at 25 °C with a cap loosely fastened to allow the solvent to evaporate sluggishly until some precipitates emerged. The precipitates were poured onto paraffin oil and were picked up by a loop.

**Computational analysis**. The FMO method[31] using the second-order Møller–Plesset perturbation theory (MP2) with the resolution-of-the-identity (RI) approximation was used to elucidate the insight into the intermolecular energy between contact pairs of **Py₆Mes**. Firstly, each molecule of **Py₆Mes** was divided into four molecular fragments: F1, F2, F3 (1,3-di(pyridin-4-yl)benzene), and F4 (mesitylene) as shown in Supplementary Fig. 23c. The geometry optimization was then performed using the standard 6–31 + G(d) basis set implemented in GAMESS program package[32]. The molecular coordinates remained the same as the initial structure during the FMO calculation. Among the eight fragments of the contact pairs of **Py₆Mes** (Supplementary Fig. 23a, b), any two fragments (*I* and *J*) were subjected to the calculation of the interaction energy decomposition analysis (PIEDA, Supplementary Table 15)[30]. The total of the contributed energy terms ($E^{total}$)) is given in Eq. (1).

$$E^{total} = \Delta E_{IJ}^{ES} + \Delta E_{IJ}^{CT+mix} + \Delta E_{IJ}^{vdW} + \Delta E_{IJ}^{EX} \qquad (1)$$

where $E^{ES}$ is the classical electrostatic energy between **Py₆Mes**, $E^{CT+mix}$ is the charge transfer energy with higher-order mixed terms energies, $E^{vdW}$ is the vdW dispersion energy, and $E^{EX}$ is the exchange repulsion between the adjacent fragments.

The total of the attractive energies ($E^{att}$) is given in Eq. 2.

$$E^{att} = \Delta E_{IJ}^{ES} + \Delta E_{IJ}^{CT+mix} + \Delta E_{IJ}^{vdW} \qquad (2)$$

The total system energies of **Py$^{open}$·MeCN** in a series of organic solvents with different relative permittivity $\varepsilon$ are calculated by the conductor-like polarizable continuum model method.

## Data availability

The data that support the findings in this study are available within the article and its Supplementary Information and/or from the corresponding authors on reasonable request. The X-ray crystallographic data reported in this article is deposited at the Cambridge Crystallographic Data Center (CCDC) under deposition numbers of 2072485–2072491 and 2095190–2095195. These data can be obtained free of charge from The Cambridge Crystallographic Data Center via www.ccdc.cam.ac.uk/data_request/cif.

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

## Acknowledgements

This work was supported by a Grant-in-Aid for Scientific Research on Young Scientist (JP19K15334) from Japan Society for the Promotion of Science (JSPS), Kato Foundation for Promotion of Science, and The Kao Foundation for Arts and Sciences, and the Nanotechnology Platform project from the Ministry of Education, Culture, Sports, Science, and Technology (MEXT), Japan.

## Author contributions

H.Y. designed the experiments. H.Y., M.T., and K.I. conducted the organic synthesis, crystallization, and characterizations. K.H. and Y.S. conducted the computational calculations. H.Y., M.T., and H.S. conducted single-crystal X-ray structural analysis. H.Y. and Y.Y. analyzed the data and prepared the manuscript with the feedback from the other authors.

## Competing interests

The authors declare no competing interests.
