## [Peer Review File · Communications Chemistry]

Reviewers' comments:

Reviewer #1 (Remarks to the Author):

This manuscript reported by Yamagishi et. al. describes a solvophobicity-driven strategy to assemble porous crystals and other polymorphs. Computational studies were carried out to detail the contributions from various intermolecular interactions to the assembly patterns observed in crystal packing. The authors attempt to utilize the Hansen solubility parameters to rationalize the formation patterns of various polymorphs. The plot of Hasen space for polymorphs of Py6Mes does shows the correlations between the solvent Hasen parameters and the resulting polymorphs. A similar correlation was also observed for the isostructural m-Py6Mes. While interesting, the current work is somewhat an extension of the previous report (reference 18) by the same group. The generality of the major findings in this work regarding the correlations between the solvent parameters and the resulting polymorphs is not very convincing because there are only two molecular examples. The results discussed are like to be case studies. This manuscript is perhaps more suitable to be published in some journals specialized in crystal engineering studies rather than Nature Communications. I do not recommend acceptance.

Reviewer #2 (Remarks to the Author):

It is surely interesting work. I recommend publication of this work in Communications Chemistry. With data quality and well considered discussions, it can be even published without serious corrections and revisions. I have some general requests and collections.

1) The results were well classified with Hansen solubility parameters. Although it is a nice strategy to interpret the obtained tendencies, it may have some difference from energy-related discussions on solvophobicity such as total system energies with solvents of crystalline energy in certain dielectric conditions. The current interpretation is OK by does not satisfy general understanding of mechanisms. Possibly, please add some descriptions on that points.

2) Solvent-dependent formation of crystal structures with pores are known in old works by Yasuhiro Aoyama for organic zeolite crystals with hydrogen bonding and Mikiji Miyata for steroid crystals. MOF and recent materials may not be most appropriate substances for comparisons.

Reviewer #3 (Remarks to the Author):

Manuscript Title: Solvophobicity-directed Assembly into Microporous Molecular Crystal

Corresponding Author: Prof. Hiroshi Yamagishi

Recommendation: Minor Revision.

Additional Comments: The manuscript entitled "Solvophobicity-directed Assembly into Microporous Molecular Crystal" by Prof. Hiroshi Yamagishi describes a solvophobicity effect in the self-assembly of discrete molecules Py6Mes into a porous form as exemplified by the crystal growth in different solvents. Furthermore, a systematical computational analysis of the crystal structures have been

carried out to elucidating the assembling mechanism. The results are interesting and useful in the field of porous organic crystals. However, only one point should be addressed before the final acceptance in Communications Chemistry. Please provide the gas sorption data of the porous samples in the manuscript.

Point-by-Point **Response** to Referees' Comments

Response to the Comments by Reviewer #1

1. This manuscript reported by Yamagishi et. al. describes a solvophobicity-driven strategy to assemble porous crystals and other polymorphs. Computational studies were carried out to detail the contributions from various intermolecular interactions to the assembly patterns observed in crystal packing. The authors attempt to utilize the Hansen solubility parameters to rationalize the formation patterns of various polymorphs. The plot of Hansen space for polymorphs of Py₆Mes does shows the correlations between the solvent Hansen parameters and the resulting polymorphs. A similar correlation was also observed for the isostructural m-Py₆Mes. While interesting, the current work is somewhat an extension of the previous report (reference 18) by the same group. The generality of the major findings in this work regarding the correlations between the solvent parameters and the resulting polymorphs is not very convincing because there are only two molecular examples. The results discussed are like to be case studies. This manuscript is perhaps more suitable to be published in some journals specialized in crystal engineering studies rather than Nature Communications. I do not recommend acceptance.

=> We appreciate the reviewer's feedback on our manuscript especially in regard to the novelty and generality. To address the first issue, we added to the introductory part some sentences about the difference between the present work and our previous report (**Revision A**). Besides, we newly synthesized and analysed a new **Py₆Mes**-analogue, **Ph₆Mes**, and its polymorphs to enhance the generality of the proposed design strategy (**Revision B**).

=> **Revision A:** Polarity (or dielectric constant) has been known as the primal parameter that dominates the polymorphism of organic molecules. In the previous study (ref 18), we analysed 4 polymorphs of **Py₆Mes** based on this conventional understanding and reported a plausible effect of the solvent polarity on the polymorphism. In contrast, in the present study, we conducted structural analysis of newly synthesized 13 polymorphs of **Py₆Mes**, 4 polymorphs of *m*-**Py₆Mes**, and 6 polymorphs of **Ph₆Mes** (*vide infra*) with the aim to clarify the above-mentioned preconception and, fortunately, succeeded in finding a new physical parameter, Hansen parameter, as a more appropriate descriptor for the polymorphism. In our opinion, this is an informative finding rather than a simple case study.

For enhancing the readability especially as to the novelty, we revised the manuscript as follows (page 4, line 13): "Along this line, we also reported, in the previous report, a

plausible molecular assembly mechanism for Py^{open} based on its four types of polymorphs.¹⁸ However, the available crystallographic data were limited at that period and, thus, we could not establish a reliable and general design strategy toward porous molecular crystals.”

=> **Revision B:** To verify the generality of our statement about Hansen parameters, we newly synthesized a novel **Py₆Mes**-analogue, **Ph₆Mes** (Figure 5a), which is a D_{3h} -symmetric molecule featuring three aromatic blades at the periphery of the mesitylene core. Instead of the six outermost pyridyl rings in **Py₆Mes**, **Ph₆Mes** features six phenyl rings. We crystallized **Ph₆Mes** in a series of organic solvents and succeeded in analysing single-crystal X-ray structures of its polymorphs in MeCN, EtOAc, THF, CHCl₃, toluene, and CH₂Cl₂. As shown in the newly added Figure 5, we observed solvophobicity-dependent polymorphism that is analogous to the polymorphism of **Py₆Mes** and *m*-**Py₆Mes**. When crystallized in MeCN featuring low δ_D value, **Ph₆Mes** underwent the exclusive crystallization without uptaking guest solvent molecules into the crystal. In contrast, when crystallized in EtOAc, THF, CHCl₃, toluene, and CH₂Cl₂ featuring δ_D values larger than MeCN, **Ph₆Mes** forms inclusion crystals with the solvent molecules. This result supports our statement about the solvophobicity-driven polymorphism and, thus, enhancing its generality. The experimental details and thus obtained molecular insight are written in the revised manuscript as follows (page 12, line 6): “*Ph₆Mes is analogously crystallized in MeCN, EtOAc, THF, CHCl₃, toluene and CH₂Cl₂, successfully yielding diffraction-quality single-crystals, whose crystal packing diagrams and crystal structure information are shown in Figure 5b–g and Tables 3 and S8–S13, respectively. In analogy with m-Py₆Mes, non-porous inclusion crystals are obtained when solvents with large δ_D (EtOAc, THF, CHCl₃, toluene and CH₂Cl₂) are utilized (Figures 5c–g, S17–S21), while crystals from MeCN include no guest solvent molecules (Figures 5b and S16).*”

Response to the Comments by Reviewer #2

1. It is surely interesting work. I recommend publication of this work in Communications Chemistry. With data quality and well considered discussions, it can be even published without serious corrections and revisions. I have some general requests and collections.

=> We sincerely thank the reviewer for the high evaluation on our work.

2. 1) The results were well classified with Hansen solubility parameters. Although it is a nice strategy to interpret the obtained tendencies, it may have some difference from energy-related discussions on solvophobicity such as total system energies with solvents of crystalline energy in certain dielectric conditions. The current interpretation is OK by does not satisfy general understanding of mechanisms. Possibly, please add some descriptions on that points.

=> We appreciate the reviewer's insightful comment. We agree with the reviewer's opinion that the crystallization of organic molecules is energetically and molecularly complicated event that cannot be described simply by the Hansen solubility parameters. To address this issue, we newly conducted computational investigations into how the dielectric constant of the solvent affects the total system energies of the porous architecture of **Py^{open}**. Methanol, chloroform, acetone, toluene, and dichloroethane, which are available in the Gamess program, were chosen as the matrix for the calculation. The calculated total system energies of the porous architecture (Table S16) exhibit a positive monotonic relationship with the dielectric constant of the medium, although the energetic contribution from the surrounding environment is relatively small. This tendency is in line with the conventional understanding that the polarization is stabilized in highly polar media. Based on these results, we concluded that not only the solvophobicity but also the polarity of the solvent, in part, facilitates the formation of porous molecular crystals. Accordingly, we revised the main manuscript as follows (page 6, line 8): *“Subsequently, we conducted computational investigation into the effect of polarity of the crystallization solvents, which has been considered as an essential parameter for predicting the polymorphism. We calculate the total system energy of **Py^{open}** on the assumption that the constituent **Py₆Mes** molecules are surrounded by MeOH, CHCl₃, acetone, toluene, and dichloroethane, respectively, which are available in Gamess program. As summarized in Table 16, the porous architecture is stabilized more as the polarity of the surrounding solvent increases, yet the change in stabilization energy from the surrounding environment estimated by the PCM method is relatively*

*small in comparison with the energetic gain from dispersion force. Overall, the porous assembly of **Py₆Mes** is sustained dominantly by the dispersion forces together with the stabilization by the polarity of the surrounding media.”* We also added some words to the conclusion of the revised manuscript as follows (page 13, line 5): “, which is further stabilized by the polarity of the solvent.”

3. 2) Solvent-dependent formation of crystal structures with pores are known in old works by Yasuhiro Aoyama for organic zeolite crystals with hydrogen bonding and Mikiji Miyata for steroid crystals. MOF and recent materials may not be most appropriate substances for comparisons.

=> According to the reviewer’s comment, we add, to the main manuscript, references (21–24) describing the solvent-dependent polymorphism of molecular crystals. We also added to the manuscript some descriptions about the organic zeolites as follows (page 3, line 11): “*Organic zeolites are a well-known class of such compounds that can uptake/release guest solvent molecules efficiently and selectively depending on the geometry and chemical affinity, yet organic zeolites are not truly porous materials because their pores readily collapse upon removing the guests.*^{21–24} *More recently, several organic crystals that can retain vacant pores have been developed.*^{9–20,}”

Response to the Comments by Reviewer #3

1. **Recommendation:** Minor Revision.

Additional Comments: The manuscript entitled “Solvophobicity-directed Assembly into Microporous Molecular Crystal” by Prof. Hiroshi Yamagishi describes a solvophobicity effect in the self-assembly of discrete molecules Py_6Mes into a porous form as exemplified by the crystal growth in different solvents. Furthermore, a systematical computational analysis of the crystal structures have been carried out to elucidating the assembling mechanism. The results are interesting and useful in the field of porous organic crystals.

=> We sincerely thank the reviewer for the high evaluation on our work.

2. However, only one point should be addressed before the final acceptance in Communications Chemistry. Please provide the gas sorption data of the porous samples in the manuscript.

=> We appreciate the reviewer's comment. We conducted a N_2 gas sorption measurement at 77 K with dried powder specimen of $\text{Py}^{\text{open}}\cdot\text{EtOAc}$ as shown in the newly added Figure S22. As expected from its porous architecture, a type I profile with a BET surface area of $597 \text{ cm}^2 \text{ mol}^{-1}$ is observed. Accordingly, we added to the revised manuscript (page 7, line 12) a description about the sorption behaviour as follows: “*Pore size distribution of $\text{Py}^{\text{open}}\cdot\text{EtOAc}$ calculated from its N_2 adsorption isotherm profile (Figure S22) is nearly identical with that of $\text{Py}^{\text{open}}\cdot\text{MeCN}$,¹⁸ while its BET surface area ($597 \text{ m}^2 \text{ g}^{-1}$) is larger than $\text{Py}^{\text{open}}\cdot\text{MeCN}$ plausibly due to the higher structural integrity of $\text{Py}^{\text{open}}\cdot\text{EtOAc}$ crystals.*”

Amended Figures

Amended Figure 5. | Crystal packing diagrams of the polymorphs of Ph_6Mes . a, Molecular structure of Ph_6Mes . b–g. Crystal packing diagrams of $\text{Ph}^{\text{VDW}}\cdot\text{MeCN}$ (b), $\text{Ph}^{\text{VDW}}\cdot\text{EtOAc}$ (c), $\text{Ph}^{\text{VDW}}\cdot\text{THF}$ (d), $\text{Ph}^{\text{VDW}}\cdot\text{CHCl}_3$ (e), $\text{Ph}^{\text{VDW}}\cdot\text{C}_7\text{H}_8$ (f), and $\text{Ph}^{\text{VDW}}\cdot\text{CH}_2\text{Cl}_2$ (g). The guest toluene molecules are colored in orange for clarity.

Amended Table 3. | Crystallographic information of the polymorphs of Ph_6Mes . Crystal structure information of $\text{Ph}^{\text{VDW}}\cdot\text{MeCN}$, $\text{Ph}^{\text{VDW}}\cdot\text{EtOAc}$, $\text{Ph}^{\text{VDW}}\cdot\text{THF}$, $\text{Ph}^{\text{VDW}}\cdot\text{CHCl}_3$, $\text{Ph}^{\text{VDW}}\cdot\text{C}_7\text{H}_8$, and $\text{Ph}^{\text{VDW}}\cdot\text{CH}_2\text{Cl}_2$ together with electric permittivity (ϵ)³² and the Hansen dispersion cohesion parameters (δ_{D})³³ of the crystallization solvents. The non-porous crystal and inclusion crystals are respectively colored in red and blue.

Crystallization Solvent	Symbol	ϵ	δ_{D} (MPa ^{0.5})	Space Group	Cell Volume (Å ³)	Volume per Py_6Mes (Å ³)
MeCN	$\text{Ph}^{\text{VDW}}\cdot\text{MeCN}$	37.5	15.3	P -1	4348	1087
EtOAc	$\text{Ph}^{\text{VDW}}\cdot\text{EtOAc}$	6.02	15.8	P -1	2475	1238
THF	$\text{Ph}^{\text{VDW}}\cdot\text{THF}$	7.52	16.8	C 2/ c	5207	1302
CHCl_3	$\text{Ph}^{\text{VDW}}\cdot\text{CHCl}_3$	4.81	17.8	C 2/ c	5209	1302
Toluene	$\text{Ph}^{\text{VDW}}\cdot\text{C}_7\text{H}_8$	2.38	18.0	P -1	2471	1236
CH_2Cl_2	$\text{Ph}^{\text{VDW}}\cdot\text{CH}_2\text{Cl}_2$	9.08	18.2	P -1	2464	1232

Amended Figure S5. ^1H NMR spectrum of **Ph₆Mes** (CDCl_3 , 400 MHz).

Amended Figure S6. ^{13}C NMR spectrum of **Ph₆Mes** (CDCl_3 , 100 MHz).

Amended Table S8. Crystal structure information of **Ph^{VDW}•MeCN**.

Crystal data	
Chemical formula	C ₆₃ H ₄₈
M_r	805.01
Crystal system, space group	Triclinic, P $\bar{1}$
Temperature (K)	100
a , b , c (Å)	10.1054 (1), 11.3531 (2), 40.4637 (4)
α , β , γ (°)	93.305 (1), 90.584 (1), 110.178 (1)
V (Å ³)	4347.90 (10)
Z	4
Radiation type	Cu K α
μ (mm ⁻¹)	0.52
Crystal size (mm)	0.50 × 0.10 × 0.02
Data collection	
Diffractometer	Rigaku XtaLAB Synergy-Custom
Absorption correction	Multi-scan CrysAlis PRO 1.171.41.110a (Rigaku Oxford Diffraction, 2021) Empirical absorption correction using spherical harmonics, implemented in SCALE3 ABSPACK scaling algorithm.
T_{min} , T_{max}	0.446, 1.000
No. of measured, independent and observed [I > 2 σ (I)] reflections	55079, 17073, 14291
R_{int}	0.044
($\sin \theta/\lambda$) _{max} (Å ⁻¹)	0.626
Refinement	
R [F ² > 2 σ (F ²)], wR (F ²), S	0.047, 0.131, 1.05
No. of reflections	17073
No. of parameters	1141
H-atom treatment	H-atom parameters constrained
$\Delta\rho_{\text{max}}$, $\Delta\rho_{\text{min}}$ (e Å ⁻³)	0.29, -0.34

Computer programs: *CrysAlis PRO* 1.171.41.110a (Rigaku OD, 2021), SHELXT (Sheldrick, 2015), *SHELXL* 2018/3 (Sheldrick, 2015), *OLEX2* 1.3 (Dolomanov *et al.*, 2009).

References

- Dolomanov, O. V., Bourhis, L. J., Gildea, R. J., Howard, J. A. K. & Puschmann, H. (2009). *J. Appl. Cryst.* **42**, 339–341.
- Sheldrick, G. M. (2015). *Acta Cryst.* **A71**, 3–8.
- Sheldrick, G. M. (2015). *Acta Cryst.* **C71**, 3–8.

Amended Figure S16. An ORTEP diagram of $\text{Ph}^{\text{VDW}} \cdot \text{MeCN}$ with a probability level of 50 %.

Amended Table S9. Crystal structure information of **Ph^{VDW}•EtOAc**.

Crystal data	
Chemical formula	C ₆₃ H ₄₈ •C ₄ H ₈ O ₂
M _r	893.11
Crystal system, space group	Triclinic, P $\bar{1}$
Temperature (K)	100
a , b , c (Å)	11.2338 (2), 14.5022 (3), 17.3472 (3)
α , β , γ (°)	70.282 (2), 80.554 (2), 68.644 (2)
V (Å ³)	2474.99 (9)
Z	2
Radiation type	Cu K α
μ (mm ⁻¹)	0.54
Crystal size (mm)	0.45 × 0.28 × 0.22
Data collection	
Diffractometer	Rigaku XtaLAB Synergy-Custom
Absorption correction	Multi-scan CrysAlis PRO 1.171.41.110a (Rigaku Oxford Diffraction, 2021) Empirical absorption correction using spherical harmonics, implemented in SCALE3 ABSPACK scaling algorithm.
T _{min} , T _{max}	0.649, 1.000
No. of measured, independent and observed [I > 2 σ (I)] reflections	28171, 9732, 8656
R _{int}	0.065
($\sin \theta/\lambda$) _{max} (Å ⁻¹)	0.626
Refinement	
R [F ² > 2 σ (F ²)], wR (F ²), S	0.054, 0.160, 1.05
No. of reflections	9732
No. of parameters	627
H-atom treatment	H-atom parameters constrained
$\Delta\rho_{\max}$, $\Delta\rho_{\min}$ (e Å ⁻³)	0.30, -0.42

Computer programs: *CrysAlis PRO* 1.171.41.110a (Rigaku OD, 2021), SHELXT 2018/2 (Sheldrick, 2018), SHELXL 2018/3 (Sheldrick, 2015), OLEX2 1.3 (Dolomanov *et al.*, 2009).

References

- Dolomanov, O. V., Bourhis, L. J., Gildea, R. J., Howard, J. A. K. & Puschmann, H. (2009). *J. Appl. Cryst.* **42**, 339–341.
- Sheldrick, G. M. (2015). *Acta Cryst.* **A71**, 3–8.
- Sheldrick, G. M. (2015). *Acta Cryst.* **C71**, 3–8.

Amended Figure S17. An ORTEP diagram of **Ph^{VDW} • EtOAc** with a probability level of 50 %.

Amended Table S10. Crystal structure information of **Ph^{VDW}•THF**.

Crystal data	
Chemical formula	2(C ₄ H ₈ O)•C ₆₃ H ₄₈
M_r	949.22
Crystal system, space group	Monoclinic, C2/c
Temperature (K)	100
a , b , c (Å)	16.8279 (3), 12.9634 (2), 24.2308 (4)
β (°)	99.885 (2)
V (Å ³)	5207.40 (15)
Z	4
Radiation type	Cu K α
μ (mm ⁻¹)	0.54
Crystal size (mm)	0.37 × 0.31 × 0.13
Data collection	
Diffractometer	Rigaku XtaLAB Synergy-Custom
Absorption correction	Multi-scan CrysAlis PRO 1.171.41.110a (Rigaku Oxford Diffraction, 2021) Empirical absorption correction using spherical harmonics, implemented in SCALE3 ABSPACK scaling algorithm.
T_{min} , T_{max}	0.568, 1.000
No. of measured, independent and observed [I > 2σ(I)] reflections	16182, 5093, 4737
R_{int}	0.021
(sin θ/λ) _{max} (Å ⁻¹)	0.626
Refinement	
R [F ² > 2σ(F ²)], wR (F ²), S	0.042, 0.111, 1.05
No. of reflections	5093
No. of parameters	345
H-atom treatment	H-atom parameters constrained
Δρ _{max} , Δρ _{min} (e Å ⁻³)	0.40, -0.33

Computer programs: *CrysAlis PRO* 1.171.41.110a (Rigaku OD, 2021), SHELXT (Sheldrick, 2015), *SHELXL* 2018/3 (Sheldrick, 2015), *OLEX2* 1.3 (Dolomanov *et al.*, 2009).

References

- Dolomanov, O. V., Bourhis, L. J., Gildea, R. J., Howard, J. A. K. & Puschmann, H. (2009). *J. Appl. Cryst.* **42**, 339–341.
- Sheldrick, G. M. (2015). *Acta Cryst.* **A71**, 3–8.
- Sheldrick, G. M. (2015). *Acta Cryst.* **C71**, 3–8.

Amended Figure S18. An ORTEP diagram of **Ph^{VDW}•THF** with a probability level of 50 %.

Amended Table S11. Crystal structure information of $\text{Ph}^{\text{VDW}} \cdot \text{CHCl}_3$.

Crystal data	
Chemical formula	$2(\text{CHCl}_3) \cdot \text{C}_{63}\text{H}_{48}$
M_r	1043.75
Crystal system, space group	Monoclinic, $C2/c$
Temperature (K)	100
a, b, c (Å)	16.9997 (4), 13.1549 (3), 23.5266 (5)
β (°)	98.076 (2)
V (Å ³)	5209.1 (2)
Z	4
Radiation type	Cu $K\alpha$
μ (mm ⁻¹)	3.33
Crystal size (mm)	$0.67 \times 0.56 \times 0.28$
Data collection	
Diffractometer	Rigaku XtaLAB Synergy-Custom
Absorption correction	Gaussian CrysAlis PRO 1.171.41.110a (Rigaku Oxford Diffraction, 2021) Numerical absorption correction based on gaussian integration over a multifaceted crystal model Empirical absorption correction using spherical harmonics, implemented in SCALE3 ABSPACK scaling algorithm.
T_{\min}, T_{\max}	0.035, 1.000
No. of measured, independent and observed [$I > 2\sigma(I)$] reflections	14749, 5082, 4638
R_{int}	0.079
$(\sin \theta/\lambda)_{\text{max}}$ (Å ⁻¹)	0.626
Refinement	
$R[F^2 > 2\sigma(F^2)], wR(F^2), S$	0.090, 0.247, 1.03
No. of reflections	5082
No. of parameters	335
H-atom treatment	H-atom parameters constrained
$\Delta\rho_{\text{max}}, \Delta\rho_{\text{min}}$ (e Å ⁻³)	0.89, -0.66

Computer programs: *CrysAlis PRO* 1.171.41.110a (Rigaku OD, 2021), SHELXT (Sheldrick, 2015), SHELXL 2018/3 (Sheldrick, 2015), OLEX2 1.3 (Dolomanov *et al.*, 2009).

References

- Dolomanov, O. V., Bourhis, L. J., Gildea, R. J., Howard, J. A. K. & Puschmann, H. (2009). *J. Appl. Cryst.* **42**, 339–341.
- Sheldrick, G. M. (2015). *Acta Cryst.* **A71**, 3–8.
- Sheldrick, G. M. (2015). *Acta Cryst.* **C71**, 3–8.

Amended Figure S19. An ORTEP diagram of **Ph^{VDW} • CHCl₃** with a probability level of 50 %.

Amended Table S12. Crystal structure information of $\text{Ph}^{\text{VDW}} \cdot \text{C}_7\text{H}_8$.

Crystal data	
Chemical formula	$\text{C}_{63}\text{H}_{48} \cdot \text{C}_7\text{H}_8$
M_r	897.14
Crystal system, space group	Triclinic, $P\bar{1}$
Temperature (K)	100
a, b, c (Å)	11.2539 (3), 14.5508 (3), 17.3425 (4)
α, β, γ (°)	70.210 (2), 78.856 (2), 68.043 (2)
V (Å ³)	2471.10 (11)
Z	2
Radiation type	Cu $K\alpha$
μ (mm ⁻¹)	0.51
Crystal size (mm)	0.24 × 0.16 × 0.10
Data collection	
Diffractionmeter	Rigaku XtaLAB Synergy-Custom
Absorption correction	Multi-scan CrysAlis PRO 1.171.41.110a (Rigaku Oxford Diffraction, 2021) Empirical absorption correction using spherical harmonics, implemented in SCALE3 ABSPACK scaling algorithm.
$T_{\text{min}}, T_{\text{max}}$	0.789, 1.000
No. of measured, independent and observed [$I > 2\sigma(I)$] reflections	29336, 9617, 8981
R_{int}	0.017
$(\sin \theta/\lambda)_{\text{max}}$ (Å ⁻¹)	0.626
Refinement	
$R[F^2 > 2\sigma(F^2)], wR(F^2), S$	0.045, 0.130, 1.03
No. of reflections	9617
No. of parameters	722
No. of restraints	126
H-atom treatment	H-atom parameters constrained
$\Delta\rho_{\text{max}}, \Delta\rho_{\text{min}}$ (e Å ⁻³)	0.33, -0.40

Computer programs: *CrysAlis PRO* 1.171.41.110a (Rigaku OD, 2021), SHELXT (Sheldrick, 2015), SHELXL 2018/3 (Sheldrick, 2015), OLEX2 1.3 (Dolomanov *et al.*, 2009).

References

- Dolomanov, O. V., Bourhis, L. J., Gildea, R. J., Howard, J. A. K. & Puschmann, H. (2009). *J. Appl. Cryst.* **42**, 339–341.
- Sheldrick, G. M. (2015). *Acta Cryst.* **A71**, 3–8.
- Sheldrick, G. M. (2015). *Acta Cryst.* **C71**, 3–8.

Amended Figure S20. An ORTEP diagram of $\text{Ph}^{\text{VDW}} \cdot \text{C}_7\text{H}_8$ with a probability level of 50 %.

Amended Table S13. Crystal structure information of $\text{Ph}^{\text{VDW}} \cdot \text{CH}_2\text{Cl}_2$.

Crystal data	
Chemical formula	$\text{C}_{63}\text{H}_{48} \cdot 1(\text{CH}_2\text{Cl}_2)$
M_r	889.94
Crystal system, space group	Triclinic, $P\bar{1}$
Temperature (K)	100
a, b, c (Å)	11.2178 (3), 14.6142 (4), 17.3807 (3)
α, β, γ (°)	69.653 (2), 78.471 (2), 67.667 (2)
V (Å ³)	2463.65 (11)
Z	2
Radiation type	Cu $K\alpha$
μ (mm ⁻¹)	1.48
Crystal size (mm)	0.22 × 0.05 × 0.03
Data collection	
Diffractometer	Rigaku XtaLAB Synergy-Custom
Absorption correction	Multi-scan CrysAlis PRO 1.171.41.110a (Rigaku Oxford Diffraction, 2021) Empirical absorption correction using spherical harmonics, implemented in SCALE3 ABSPACK scaling algorithm.
$T_{\text{min}}, T_{\text{max}}$	0.738, 1.000
No. of measured, independent and observed [$I > 2\sigma(I)$] reflections	28682, 9768, 8526
R_{int}	0.035
$(\sin \theta/\lambda)_{\text{max}}$ (Å ⁻¹)	0.626
Refinement	
$R[F^2 > 2\sigma(F^2)], wR(F^2), S$	0.056, 0.150, 1.04
No. of reflections	9768
No. of parameters	653
No. of restraints	81
H-atom treatment	H-atom parameters constrained
$\Delta\rho_{\text{max}}, \Delta\rho_{\text{min}}$ (e Å ⁻³)	1.01, -0.50

Computer programs: *CrysAlis PRO* 1.171.41.110a (Rigaku OD, 2021), SHELXT (Sheldrick, 2015), SHELXL 2018/3 (Sheldrick, 2015), OLEX2 1.3 (Dolomanov *et al.*, 2009).

References

- Dolomanov, O. V., Bourhis, L. J., Gildea, R. J., Howard, J. A. K. & Puschmann, H. (2009). *J. Appl. Cryst.* **42**, 339–341.
- Sheldrick, G. M. (2015). *Acta Cryst.* **A71**, 3–8.
- Sheldrick, G. M. (2015). *Acta Cryst.* **C71**, 3–8.

Amended Figure S21. An ORTEP diagram of $\text{Ph}^{\text{VDW}} \cdot \text{CH}_2\text{Cl}_2$ with a probability level of 50 %.

Amended Figure S22. N_2 adsorption (black circles) and desorption (red circles) isotherms of $\text{Py}^{\text{open}}\cdot\text{EtOAc}$ measured at 77 K. The BET surface area of $\text{Py}^{\text{open}}\cdot\text{EtOAc}$ calculated based on the isotherm is $597 \text{ cm}^2 \text{ mol}^{-1}$. Inset represents the pore size distribution of $\text{Py}^{\text{open}}\cdot\text{EtOAc}$ calculated by means of micropore analysis (MP) method.

Table S16. Summary of calculated total system energies of $\text{Py}^{\text{open}}\cdot\text{MeCN}$ in a series of organic solvents with different dielectric constant ϵ .

Solvent	Vacuum	Toluene	CHCl_3	$\text{C}_2\text{H}_4\text{Cl}_2$	Acetone	MeOH
ϵ	1	2.38	4.81	10.37	20.56	32.66
Total system energy (kcal/mol)	-94.303	-101.54	-101.723	-101.904	-102.007	-102.047

REVIEWERS' COMMENTS:

Reviewer #2 (Remarks to the Author):

Replies and revisions are fine. The revised version becomes acceptable. It is a nice work.

Reviewer #3 (Remarks to the Author):

The revised version of manuscript is satisfactory. As a result, I suggest its publication.

Point-by-Point Response to Referees' Comments

Response to the Comments by Reviewer #2

1. Replies and revisions are fine. The revised version becomes acceptable. It is a nice work.

=> We appreciate the reviewer's feedback.

Response to the Comments by Reviewer #3

2. The revised version of manuscript is satisfactory. As a result, I suggest its publication.

=> We appreciate the reviewer's feedback.